# Effects of Embryonic Thermal Manipulation on Body Performance and Cecum Microbiome in Broiler Chickens Following a Post-Hatch Lipopolysaccharide Challenge

**DOI:** 10.3390/ani15081149

**Published:** 2025-04-17

**Authors:** Seif Hundam, Mohammad Borhan Al-Zghoul, Mustafa Ababneh, Lo’ai Alanagreh, Rahmeh Dahadha, Mohammad Mayyas, Daoud Alghizzawi, Minas A. Mustafa, David E. Gerrard, Rami A. Dalloul

**Affiliations:** 1Department of Basic Medical Veterinary Sciences, Faculty of Veterinary Medicine, Jordan University of Science and Technology, Irbid 22110, Jordan; sfhundam23@vet.just.edu.jo (S.H.); ababnem@just.edu.jo (M.A.); rmdahadha23@vet.just.edu.jo (R.D.); daalghizzawi20@vet.just.edu.jo (D.A.); 2Department of Medical Laboratory Sciences, Faculty of Applied Medical Sciences, The Hashemite University, Zarqa 13133, Jordan; loai-alanagreh@hu.edu.jo (L.A.); minasahmad36@gmail.com (M.A.M.); 3Department of Medical Laboratory Sciences, Faculty of Allied Medical Sciences, Zarqa University, Zarqa 13133, Jordan; 4Department of Animal Production, Faculty of Agriculture, Jordan University of Science and Technology, Irbid 22110, Jordan; mohammad.mayyas@jsmo.gov.jo; 5School of Animal Sciences, Virginia Polytechnic Institute and State University, Blacksburg, VA 24061, USA; dgerrard@vt.edu; 6Department of Poultry Science, University of Georgia, Athens, GA 30602, USA; rami.dalloul@uga.edu

**Keywords:** broiler, thermal manipulation, microbiome, lipopolysaccharide, gut health, antimicrobial

## Abstract

Reducing reliance on antibiotic growth promoters (AGPs) is a priority to combat antimicrobial resistance (AMR). This study explores how temperature modification during egg incubation, a process called thermal manipulation (TM), can enhance broilers’ gut health and performance under lipopolysaccharide (LPS)-induced stress. Using 16S rRNA gene sequencing, we analyzed the gut microbiome and found that chicks exposed to TM had a different composition of gut bacteria, particularly by increasing beneficial bacterial taxa such as *Oscillospirales*. These microbial changes may improve the birds’ immune response to LPS-induced stress, as TM-treated birds avoided the hypothermic phase typically triggered by LPS. These findings suggest that TM could be a cost-effective and easily applicable way to improve poultry health, reduce antibiotic dependence, and support sustainable farming practices.

## 1. Introduction

Chickens are essential for meeting the rising global demand for animal protein. Specifically, in 2021, chicken meat was the most produced meat worldwide, reaching approximately 120 million tons [1]. Artificial genetic selection is an essential strategy to improve the rate of meat production, resulting in a more than 400% increase in broiler growth rate over the past fifty years. Still, it has been linked with various disadvantages in the body, including negatively affected musculoskeletal, cardiac, and immune systems [2]. Additionally, it has been found to alter the gut microbiota, likely due to a modified mucosal ecological environment [3].

Antibiotic growth promoters (AGPs) have been widely used to enhance broiler performance, increasing growth rates by 4–8% and feed efficiency by 2–5% [4]. However, the broad use of AGPs leads to residues in meat and the development of antimicrobial resistance (AMR) [5]. By 2050, AMR is predicted to cause 10 million deaths annually if proper actions are not taken to slow down the threat [6]. Consequently, finding sustainable alternatives to AGPs has become a priority.

Thermal manipulation (TM) is a highly efficient and cost-effective method for improving broiler production and health, offering a promising alternative to reduce the reliance on AGPs. TM, achieved through epigenetic modifications by altering incubation temperature and humidity during the critical period of embryogenesis, has been shown to improve post-hatch performance [7,8]. Studies demonstrate that TM promotes muscle growth, strengthens resistance to bacterial challenges, and enhances immune responses under heat stress [9,10,11,12].

Lipopolysaccharide (LPS), a Gram-negative bacterial endotoxin, causes systemic inflammation and weight loss through appetite suppression and immunological pathway changes [13]. Previous research has demonstrated that TM influences immune responses by modulating splenic cytokine expression and heat shock protein (HSP) mRNA levels in LPS-challenged broiler chickens [14]. Similarly, Shanmugasundaram et al. [15] reported that TM in Pekin duck embryos significantly reduced bursal and splenic HSP70 and IL-6 mRNA levels following LPS challenge, supporting TM’s potential role in enhancing immune resilience.

The gut microbiome, composed of bacteria, fungi, archaea, protozoa, and viruses, plays a pivotal role in nutrient digestion, immunity, and growth performance in broilers [16]. TM may enhance gut microbiome resilience, potentially mitigating LPS-induced inflammation and supporting immune function. Recent studies have demonstrated significant microbial shifts during embryonic development; however, the effects of embryonic environmental conditions, such as TM, on the microbiome remain largely underexplored [17]. For instance, while TM has been shown to alter enterocyte proliferation and brush-border membrane enzyme activity—and when combined with post-hatch baicalein supplementation, it enhances volatile fatty acid production and improves gut health by promoting beneficial microbial communities—its influence on microbial communities during LPS-induced inflammation has not been thoroughly investigated [18,19]. Additionally, some studies have investigated post-hatch challenges like LPS-induced microbial change, reporting limited shifts in microbial diversity and increased pathogenic bacteria under LPS stress [20,21,22]. Research on the interaction between TM and post-hatch LPS challenge, as well as the impact on microbiome responses, remains limited.

This study investigates TM’s potential in modulating the cecal microbiome to enhance resilience against LPS-induced intestinal stress. By modifying embryonic development, TM may induce long-term shifts in gut microbial community composition and function. This approach could offer a sustainable alternative to AGPs to improve broiler health and performance.

## 2. Materials and Methods

The Animal Care and Use Committee of Jordan University of Science and Technology (Approval Number: 347/12/4/16) reviewed and approved all experimental procedures.

### 2.1. Study Population and Incubation

Figure 1 summarizes the experimental design used in the current study. After visual inspection to exclude damaged eggs, selected eggs (53.8 ± 3.49 g) were divided into two groups: the control (CON) and TM. The eggs in the CON group were kept at a regular 37.8 °C air temperature and 56% relative humidity (RH) throughout the embryogenesis period. The eggs in the TM group were incubated under regular incubation environments, except on embryonic days 10–18, when incubated at 39 °C and 65% RH for 18 h each day, consistently, from 12:00 a.m. to 8:00 a.m. All eggs were incubated in a Masalles Mod.25-I PDS incubator (Barcelona, Spain), with candling performed on embryonic day 7 to remove infertile eggs and those containing dead embryos.

According to our earlier research, using this TM procedure under heat stress in Cobb and Hubbard breeds results in changes to the expression of heat shock proteins and inflammatory cytokines, as well as a decrease in the expression of genes linked to oxidative stress and an increase in antioxidant capacity [23,24].

### 2.2. Hatching Management and Post-Hatching Rearing

Chicks were monitored hourly on the day of hatching. At 24 h post-hatching, the chicks were feather-sexed and then transferred to the animal house at Jordan University of Science and Technology, where the field experiments were conducted. The male broiler chicks were equally distributed into two groups. Each group contained 16 caged pens (80 × 40 × 100 cm). The pen was considered an experimental unit. Chicks were fed basal diets as per the National Research Council’s recommendations, fulfilling the standard nutrient requirements (Table 1), with the method adapted from Shakouri and Malekzadeh [25].

Chicks received vaccination against Newcastle disease and infectious bronchitis. All chicks were raised under similar managerial practices. The experimental trial lasted 22 days, from 15 May to 5 June. Throughout the first week, the room temperature was kept at 33 ± 1 °C; it gradually decreased to 22 °C by the end of the third week. Throughout the field trial, the chicks received meal rations ad libitum and unlimited access to water [26].

### 2.3. LPS Challenge

On post-hatch day 19, the LPS challenge was administered. Both primary groups, TM and CON, were divided into two subgroups. The CON and TM subgroups did not have LPS injections; a second subgroup received the LPS treatment (LPS-CON and LPS-TM). On days 19, 20, and 21, the LPS subgroups (*n* = 20) received three intraperitoneal injections of LPS from *E. coli* O127 at a dose of 0.5 mg/kg. This dosing protocol, adapted from previous research by Wu et al. [27], demonstrated that subsequent LPS injections reliably induced intestinal mucosal damage, characterized by hemorrhagic lesions in the ileum and jejunum, severe villus structural damage (including epithelial fragmentation and mucosal erosion), and significant alterations in cecal microbial composition consistent with gut dysbiosis. Additionally, supporting this approach Zheng et al. [28] showed that three intraperitoneal LPS injections administered at 16, 18, and 20 days of age effectively induced oxidative stress, inflammatory responses, and intestinal morphological changes in broiler chicks.

After administering the injection, each subgroup was relocated to an experimental pen and maintained under thermoneutral conditions to ensure their physiological comfort. Eight hours post-injection, six new random chick samples were selected daily from each subgroup to measure body weight (BW) and temperature (BT). Additionally, eight hours after the last LPS injection, cecal content samples were collected for microbiome analysis. Chickens were humanely euthanized, and the ceca were aseptically excised using sterile instruments. The ceca were carefully and immediately snap-frozen on site to prevent DNA degradation. Samples were stored in a CryoCube^®^ F570 Series Ultra-Low Temperature (ULT) Freezer (Eppendorf, Hamburg, Germany) at −80 °C to preserve microbial DNA integrity until extraction. A J/K/T thermocouple meter connected to a rat rectal probe (manufactured by Kent Scientific Corporation, Torrington, CT, USA; accuracy ±0.1 °C) was used to measure the chickens’ body temperature.

### 2.4. Microbiological Analysis

This study employed Illumina-based next-generation sequencing (NGS) to investigate the effects of embryonic TM on the cecum microbiome during a post-hatch LPS challenge. The Illumina platform, widely used for 16S rRNA gene sequencing, provides a highly accurate culture-independent approach for profiling microbial communities [29]. The 16S rRNA gene benefits this study due to its variable regions, which allow for the classification of microbial taxa. This method enables detailed insights into microbial diversity and community dynamics in environmental and host-associated samples.

#### 2.4.1. DNA Isolation and Sequencing

Microbial DNA was isolated from about 250 mg of cecal contents collected from six randomly selected chicks per experimental subgroup using the DNeasy PowerSoil Pro Kit (QIAGEN, Inc., Valencia, CA, USA) according to the manufacturer’s procedures [30]. The use of 6 samples in this study was informed by recent findings in broiler chicken microbiome research. Weinroth and Oakley demonstrated that sampling 6–10 birds captures most microbial diversity, with less than 1% new amplicon sequencing variants (ASVs) added beyond this point [31].

The (V3-V4) region of the 16S rRNA gene was amplified using universal primers Bakt_341F (CCTACGGGNGGCWGCAG) and Bakt_805R (GACTACHVGGGTATCTAATC C). Macrogen Inc., Seoul, Republic of Korea, performed the sequencing commercially utilizing an Illumina MiSeq platform (Illumina, San Diego, CA, USA), following the 2 × 300 bp paired-end sequencing protocol.

#### 2.4.2. Data Analysis by Bioinformatics Tools

Figure 2 depicts the comprehensive pipeline utilized for the S16 microbiome sequencing of cecal content. QIIME 2 version 2024.8 was used for microbiome bioinformatics [32], using DADA2 for quality filtering, denoising, and amplicon sequence variant (ASV) generation. Taxonomic classification was performed with the Silva 138 reference database (99% OTUs). Rarefaction was performed with 19,940 reads per sample to standardize diversity metrics, including the Chao1, Shannon, and Bray–Curtis indices. Beta diversity was analyzed using Canonical Correspondence Analysis (CCA) on Hellinger-transformed data, implemented in the ampvis2 R package. Differences in microbial composition were tested using PERMANOVA (Permutational Multivariate Analysis of Variance) and validated by PERMDISP (permutational analysis of multivariate dispersions). Differential abundance analysis was conducted via LEfSe (LDA score > 2.0, *p* < 0.05) and STAMP for Kruskal–Wallis tests. The full bioinformatics workflow and references are detailed in Appendix A S1.

### 2.5. Statistical Analyses

BT and BW were analyzed using IBM SPSS Statistics 27.0 (IBM software, Chicago, IL, USA) and are represented as means ± SD. A 2 × 2 factorial design was employed, with random sampling for each measurement time point. A two-way ANOVA was used to compare variations in BT and BW within the same day between treatment groups. To analyze changes in BT over time, a one-way ANOVA was applied separately for each group (e.g., CON group: Day 1 vs. Day 2 vs. Day 3). In addition, BT was visualized using GraphPad Software, statistical version 10.0 (San Diego, CA, USA), while BW was visualized using Microsoft Excel. *p* values lower than 0.05 were considered significantly different.

## 3. Results

### 3.1. Effects of TM and LPS on Alpha Diversity

There were no significant differences (*p* > 0.05) in richness or diversity among the groups (Table 2), suggesting that neither TM nor LPS challenge had a direct impact on alpha diversity under the experimental conditions tested.

### 3.2. Rarefaction Curve Analysis

Figure 3 displays the rarefaction curves, which illustrate the Shannon index as a function of sequencing depth. The curves reach a plateau for all groups, indicating that sufficient sequencing depth was achieved to capture the diversity within each group.

### 3.3. Effects of TM and LPS on Beta Diversity

CCA showed a significant separation among the cecal microbiota from the different experimental groups (Figure 4). Specifically, each group formed distinct clusters, indicating unique bacterial community compositions.

Pairwise PERMANOVA analysis (Table 3) revealed significant differences between the TM and CON groups (F = 2.0033, *p* = 0.041), indicating that TM influenced microbial diversity. Similarly, a significant difference was observed between the TM and LPS-CON groups (F = 3.4913, *p* = 0.012).

In contrast, the main effect of LPS was not significant (CON vs. LPS-CON: F = 1.0491, *p* = 0.398, and TM vs. LPS-TM: F = 1.5514, *p* = 0.080). PERMDISP confirmed homogeneity of variance (*p* = 0.21).

### 3.4. Effects of TM and LPS on Cecal Microbiota: Phylum-Level Composition

At the phylum level, Firmicutes (76.2–82%) and Bacteroidota (17.4–23.3%) were the dominant bacterial phyla across all groups. Minor contributions were observed from Proteobacteria (0.2–0.5%), Actinobacteriota (0.1–0.3%), and Cyanobacteria (0.0001–0.0009%) (Figure 5a).

The LPS-CON group showed the lowest relative abundance (76.2%) of *Firmicutes*, while higher levels were observed in the CON (80%) and TM (80.1%) groups. Notably, the greatest relative abundance of *Firmicutes* (82%) occurred in the LPS-TM group. Conversely, *Bacteroidota* exhibited an inverse trend. The LPS-CON group had the highest relative abundance (23.3%), while the LPS-TM group showed the lowest (17.4%). Although these differences did not reach statistical significance (*p* > 0.05), this trend may highlight the potential of TM to modulate dominant bacterial phyla in response to immunological stress, such as LPS.

### 3.5. Effects of TM and LPS on Cecal Microbiota: Order-Level Composition

In all groups, the most prevalent bacterial orders were *Lachnospirales* (27.8–36%), *Oscillospirales* (14–40%), *Bacteroidales* (17.4–23.2%), and *Lactobacillales* (7.6–21.9%). Other bacterial orders, such as *Erysipelotrichales, Clostridia vadinBB60—group*, *Clostridia UCG_014*, *Bacillales*, *Enterobacterales*, and *Monoglobales*, were present at lower abundances (Figure 5b).

*Oscillospirales* showed statistically significant differences between the TM and LPS-TM groups compared to the LPS-CON group (Figure 6 and Figure 7b). The TM and LPS-TM groups exhibited higher relative abundances (40% and 35.6%, respectively) than the LPS-CON group (14.3%) (*p* < 0.001). These findings suggest that TM promotes the growth of *Oscillospirales*.

The relative abundance of *Lactobacillales* in the LPS-CON group was 21.9%. A statistically significant difference (*p* < 0.05) in *Lactobacillales* abundance was observed between the TM and LPS-CON groups (Figure 7a). LDA further highlighted the enrichment of *Lactobacillales* in the LPS-CON group (Figure 7b).

### 3.6. TM and LPS Challenge Effects on Body Temperature (BT)

Figure 8a represents BT changes over time (Day 1 to Day 3), following LPS injections. In the LPS-CON group, BT initially decreased before increasing significantly (*p* = 0.013). This pattern reflects the typical response to LPS, which induces a hypothermic phase followed by fever as part of the acute-phase immune response. In contrast, the LPS-TM group bypassed the hypothermic phase, exhibiting a non-significant increase in body temperature over time (*p* > 0.05).

### 3.7. TM and LPS Challenge Effects on Body Weight (BW)

The line graph (Figure 8b) illustrates BW changes over time across the experimental groups in response to LPS injections. LPS significantly reduced BW (*p* < 0.001), highlighting its negative impact on growth performance. In contrast, TM did not significantly affect BW (*p* > 0.05), suggesting that TM may not mitigate the adverse effects of LPS on growth.

## 4. Discussion

This study provides new insights into how pre-hatch environmental manipulations, specifically TM, influence post-hatch microbial communities. It is among the first to investigate the impact of TM on gut microbiota composition, particularly under immunological challenges like exposure to LPS. While TM’s effects on broiler performance and heat stress resilience have been studied, its influence on gut microbial dynamics during LPS challenge has remained underexplored until now [9,10,11].

Population diversity is a crucial feature of a microbiome community that is closely linked to its environment. A change in diversity may indicate that the environment of the animal body has changed (from a healthy to a diseased state, for instance) or that certain conditions have disrupted the environment (e.g., a shift in the immune system or usage of antibiotics). Thus, one of the main goals of microbiome community research is to analyze the diversity of gut microbiota and discover host and/or environmental factors that can change the taxonomic composition [33].

Alpha diversity, which represents within-sample diversity, can be described using measures based on community richness (number of taxonomic groups), evenness (distribution of group abundances), or a combination of both [34]. The Chao1 index provides a nonparametric estimation of taxonomic richness based on abundance, while Shannon’s index (H) combines evenness and richness to estimate taxonomic diversity [35,36].

In this study, the Chao1 index was used to assess community richness, and Shannon’s index was used to represent community diversity. TM and LPS did not significantly affect overall microbial richness or diversity. These findings align with previous studies showing limited effects of LPS on alpha diversity [20,21,22].

Beta diversity measures differences in microbial community composition between groups or samples and is often more sensitive than alpha diversity metrics for detecting such distinctions [37]. In this study, a significant difference was detected between the TM and CON groups, underscoring TM’s role in shaping microbial community dynamics rather than changing microbial richness with the Bray–Curtis dissimilarity metric, which proved particularly effective in capturing these group-specific differences, highlighting the value of beta diversity in microbiome studies.

Research suggests that developmental stages and environmental factors during embryonic development significantly influence the diversity of gut microbiota. For example, Tong and Cui [38] reported that these factors shape microbiota composition in amphibians, while de Jong and Schokker [39] demonstrated that early-life environmental conditions affect gut microbial diversity in broilers. Among these factors, incubation temperature is a key regulator of epigenetic modifications, including DNA methylation and histone modifications, which influence microbial colonization and gut health [40,41]. TM specifically alters epigenetic regulation in embryos, leading to long-term physiological effects. According to David et al. [40], exposing chicken embryos to TM (39.5 °C for 12 h per day between the 7th and 16th day of embryogenesis) modifies post-translational histone markers, particularly H3K4me3 and H3K27me3, in the hypothalamus and muscle tissues. These changes contribute to environmental memory, impacting neurodevelopmental, metabolic, and immune regulatory pathways. Such epigenetic shifts likely extend to the gastrointestinal system, influencing gut microbiota composition and function.

Furthermore, the peri-hatching period is critical for microbiota establishment, as this window determines early microbial colonization, which can have long-lasting effects on birds’ health and development [41]. Early exposure to beneficial microbes allows advantageous bacterial populations to become dominant, and this is potentially modulated by TM-induced epigenetic mechanisms. Additionally, studies on in ovo microbiome stimulation with probiotics, prebiotics, and synbiotics suggest that these microbial interventions induce significant epigenetic changes, particularly in the liver and spleen. This further supports the link between early environmental interventions and gut microbiota regulation [42,43].

Animal health and performance at all levels of growth depend on maintaining an adequate balance of gut bacteria [44]. The cecum contains the most diverse microbial populations, which are essential for energy extraction and nutrient breakdown, ultimately affecting chickens’ intestinal health and growth efficiency [45,46]. In our study, *Firmicutes* and *Bacteroidota* phyla dominated the chickens’ cecal microbiota, which aligns with previous research [47]. However, the relative abundance of the dominant phyla can vary depending on factors such as diet and farming. For instance, free-range chickens tend to have higher levels of Bacteroidetes and Proteobacteria, while broiler chickens show a dominance of Firmicutes [48].

A high *Firmicutes-to-Bacteroidota* (F/B) ratio has previously been correlated with greater body weight in chickens [49]. *Firmicutes* are associated with butyrate and propionate production, which are short-chain fatty acids (SCFAs) linked to gut health and inflammation regulation [50]. In our study, the TM-LPS group exhibited a non-significantly higher F/B ratio, suggesting a potential trend toward improved growth performance.

At the order level, *Oscillospirales* were significantly enriched in the TM groups. These SCFA-producing bacteria are considered potential next-generation probiotics due to their role in synthesizing propionate, valerate, and butyrate [51,52]. Through the fermentation of dietary fibers and starch, *Oscillospirales* generate SCFAs that contribute to various physiological benefits, such as controlling apoptosis, lipid metabolism, and inflammation, influencing host DNA epigenetics, and maintaining the integrity of the epithelial barrier [53,54]. Specifically, as one of the principal butyrate producers, *Oscillospirales* play a vital role in gut health [55]. Butyrate serves as the primary energy source for enterocyte development and function [56]. Additionally, butyrate modulates the immune response by enhancing antioxidant enzyme activity, reducing pro-inflammatory cytokines, and inducing histone hyperacetylation through histone deacetylase (HDAC) inhibition—mechanisms that collectively regulate cell proliferation and likely contribute to the improved gut morphology previously observed in TM broilers [57,58,59].

Recent findings by Gałęcka et al. [60] further highlight the ecological significance of *Oscillospirales*, demonstrating that specific members (e.g., UCG-10) may respond to environmental contaminants like microplastics in porcine microbiomes, potentially contributing to both anti-inflammatory processes and contaminant degradation. Conversely, LPS-CON birds exhibited reduced *Oscillospirales* abundance, indicating that LPS exposure may selectively suppress these beneficial microbial taxa.

The increased abundance of *Oscillospirales* in the LPS-TM group may have contributed to the birds bypassing the hypothermic phase, indicating a stronger and more adaptive immune response. This could reflect enhanced cytokine production or other immunomodulatory effects induced by TM [9,14]. In contrast, the LPS-CON group displayed significantly lower BT post-LPS injection on Day 1 compared to the other groups, suggesting an initial hypothermic phase. This phase, characterized by reduced blood pressure, decreased circulating white blood cells, and increased apoptotic leukocytes, aligns with the early immune response to LPS [61].

*Lactobacillales*, a bacterial order well known for their probiotic properties, increased in the LPS-CON group in response to LPS-induced stress [62]. This microbial shift likely reflects a compensatory mechanism to mitigate gastrointestinal stress caused by LPS. *Lactobacillales* play a crucial role in immunoregulation, as their abundance has been positively correlated with SCFA production and CD4+ T-cell modulation, both of which contribute to immune homeostasis [63,64].

Additionally, *Lactobacillales* produce lactic acid, which lowers intestinal pH and creates an unfavorable environment for pathogenic bacteria, disrupting their outer membrane and limiting their colonization [65,66]. This shift in microbial composition may represent an adaptive host–microbiota interaction, where beneficial bacteria expand to protect the host from excessive inflammatory damage. Supporting this hypothesis, Alaqil [67] reported that *Lactobacillus acidophilus* (a key member of the *Lactobacillales* order) significantly alleviated stress markers and HSP70 expression in the spleen while increasing intestinal antioxidant capacity in LPS-challenged laying hens.

Furthermore, in the LPS-TM group, the abundance of *Lactobacillales* also did not significantly increase compared to the TM group, supporting the hypothesis that LPS may promote the growth of beneficial bacteria, such as *Lactobacillales*, to preserve gut health. According to Dai and Hackmann, LPS stimulation boosted the number of lactic acid-producing bacteria [68]. They also suggested that LPS would be detrimental to other bacteria that do not produce lactate, enhancing competition among lactate-producing bacteria and promoting their growth.

The observed effects of TM on gut microbiota composition may involve the microbiota–gut–brain (MGB) axis, a bidirectional communication system linking microbial dynamics with host stress responses [69]. TM has been shown to induce hypothalamic epigenetic adaptations and alter the levels of key hormones, such as plasma corticosterone, triiodothyronine (T3), and testosterone, especially during stressful situations [40,70]. These hormonal changes, mediated through mechanisms like the hypothalamic–pituitary–adrenal (HPA) axis, likely influence gut microbial composition and resilience [71,72]. Specifically, in chicken embryos, thyroid hormones and glucocorticoids work synergistically to promote gut maturation at the end of incubation. Thyroid hormones promote cellular differentiation and trigger the creation of digestive enzymes, whereas glucocorticoids aid in the development of intestinal glucose transport [73]. Interestingly, broiler chicks fed probiotic-supplemented diets have significantly increased blood T3 levels, highlighting the interrelated pathways between endocrine control, gut health, and microbial composition [74].

Compared to non-challenged groups, the LPS challenge reduced BW, consistent with previous studies demonstrating that LPS primarily affects appetite regulation and immune response, leading to weight loss. The mammalian target of rapamycin (mTOR) signaling pathway is activated by LPS, which causes anorexia by suppressing the expression of agouti-related protein (AgRP) in the hypothalamus, reducing food intake [13]. Furthermore, LPS exposure triggers the Toll-like receptor 4 (TLR4) signaling pathway, which might result in acute-phase responses and, consequently, due to immunological activation, lower body weight and feed intake may occur [75]. Remarkably, both the LPS-TM and LPS-CON groups exhibited similar growth suppression, suggesting that the LPS challenge diminished the beneficial effect of TM. However, microbiome analysis provides insight into how TM alters the intestinal ecology, indicating that TM might promote a more robust gut microbiota capable of responding to immunological stresses such as LPS.

TM shows promising effects on muscle development and growth-related hormones, but its success depends on incubation temperature, duration, and timing [76,77,78,79,80]. Studies suggest that specific TM protocols, such as incubating eggs at 38.2–38.4 °C for 2 h daily from day 18 until hatching or at 39.5 °C for 4 h daily from days 12 to 18, can increase BW [70,81]. However, other protocols, like 39.5 °C for 3 h daily during early, late, or combined embryonic periods, showed no significant effect on post-hatch BW [82]. Similarly, exposure to 39 °C and 65% RH for 12–18 h daily from days 12 to 18 significantly increased BW, while shorter durations did not [76]. These findings underscore the importance of optimizing TM conditions for the desired outcomes.

In our study, the significant decrease in BW post-LPS in the Indian River breed suggests that TM protocols may need further optimization for this specific breed. TM responses can vary across chicken breeds or types. For instance, TM in layer-type chickens has shown less favorable outcomes compared to broilers [83]. Limited literature exists on the effects of TM on the Indian River breed, highlighting the need for additional studies to optimize TM protocol for this breed.

This study underscores the potential of thermal manipulation (TM) as a non-invasive and cost-effective approach to enhancing gut microbiota resilience against immunological stress. TM-induced microbial shifts, particularly the enrichment of SCFA-producing taxa such as *Oscillospirales*, present promising avenues for improving poultry health and reducing antibiotic dependency. While TM enhances some microbial composition, its direct impact on growth performance during inflammatory stress remains unclear. Future research should prioritize metagenomic and metabolomic analyses to elucidate the functional pathways involved in TM-modulated gut microbiota. Such studies could provide a clearer understanding of TM’s potential in mitigating LPS-induced growth suppression. Integrating TM with nutritional strategies, including prebiotics, probiotics, phytogenic feed additives, and organic acids, offers a comprehensive solution to reduce antibiotic utilization and mitigate the risk of antimicrobial resistance (AMR).

## 5. Conclusions

The findings of this investigation demonstrate that thermal manipulation during embryogenesis dynamically and persistently modifies the gut microbiota. Moreover, by increasing beneficial bacteria and avoiding the hypothermic phase, TM significantly enhanced broiler response to the LPS challenge. These findings support TM as a viable antibiotic-free strategy for commercial poultry production. Future work should optimize protocols for different strains and integrate probiotic approaches. Standardized hatchery implementation could improve flock health while reducing antibiotic use, addressing critical industry needs.

## 6. Study Limitations

This study has several limitations that should be considered. First, the exclusive use of male chicks due to budget constraints may affect the generalizability of our findings, as established sex differences in broiler microbiome composition and function are well-documented. Second, while we focused on post-LPS microbiome changes, the characterization of microbial dynamics during the early post-hatch period (days 1–18) before the LPS challenge would have provided valuable insights into how TM shapes initial microbial colonization patterns. Additionally, although our findings suggest a potential link between TM, immune modulation, and metabolic efficiency, we did not directly measure muscle mass, carcass yield, or SCFA production. Future studies should incorporate muscle fiber histology, carcass composition analysis, metabolomic profiling, multi-timepoint microbiome analysis, and functional metagenomics to establish a more direct relationship between TM, microbial modulation, and overall broiler performance.

## Figures and Tables

**Figure 1 animals-15-01149-f001:**
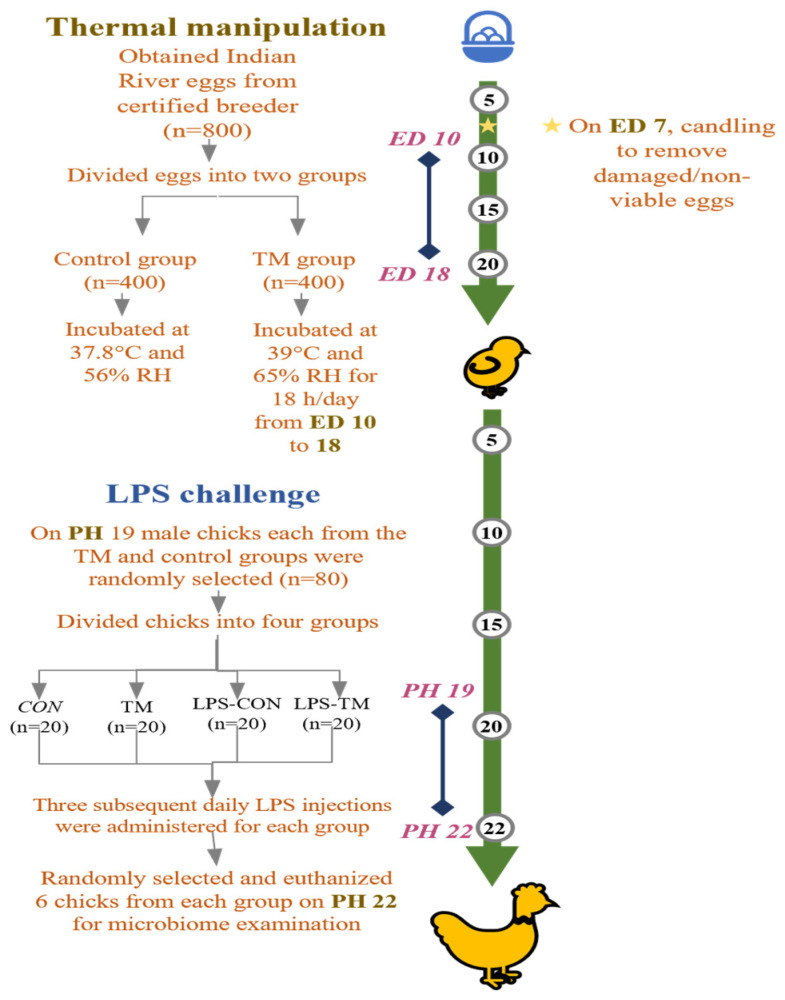
Summary of the experimental design carried out in this study. TM: thermal manipulation, CON: control, RH: relative humidity, LPS: lipopolysaccharide, ED: embryonic day, PH: post-hatch.

**Figure 2 animals-15-01149-f002:**
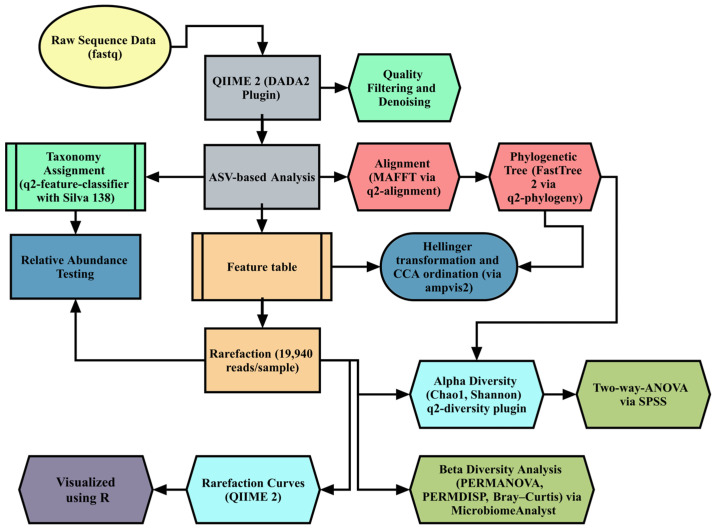
Summary of the bioinformatics analysis carried out in this study. ASV: amplicon sequencing variant; CCA: canonical correspondence analysis; PERMANOVA: Permutational Multivariate Analysis of Variance; PERMDISP: permutational analysis of multivariate dispersions.

**Figure 3 animals-15-01149-f003:**
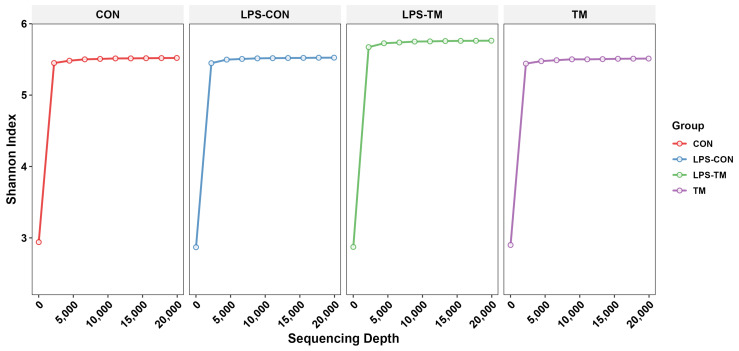
The curves approach a plateau, indicating that the sequencing depth was sufficient to capture the most microbial diversity in all groups.

**Figure 4 animals-15-01149-f004:**
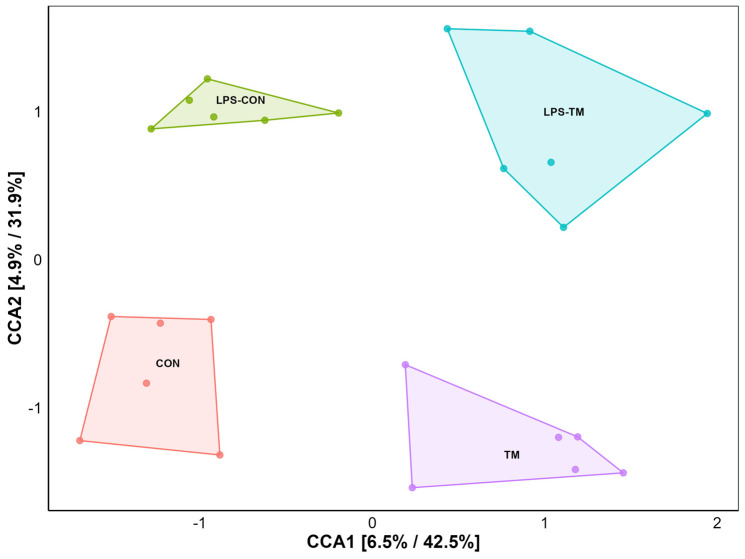
Ordination plot for beta diversity generated by canonical correspondence analysis (CCA). The CCA plot illustrates the similarity between microbial communities based on beta diversity. Each point represents a sample.

**Figure 5 animals-15-01149-f005:**
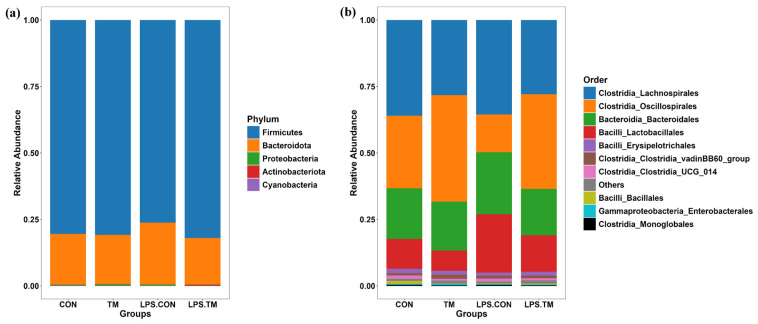
Cecum microbiota composition at the phylum and order levels. (**a**) Composition at the phylum level. No significant differences were observed among groups. (**b**) Composition at the order level. *Oscillospirales* and *Lactobacillales* showed significantly different relative abundances (*p* < 0.05) between groups.

**Figure 6 animals-15-01149-f006:**
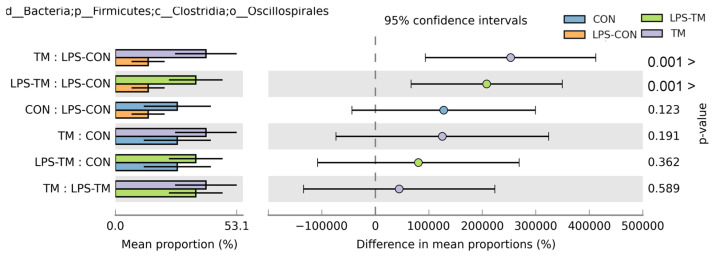
Relative abundance of *Oscillospirales* across experimental groups. Results of the Kruskal–Wallis H-test indicate statistically significant differences (*p* < 0.05) in the relative abundance of *Oscillospirales* among the TM, LPS-TM, and LPS-CON groups. The TM and LPS-TM groups exhibited higher relative abundances (40% and 35.6%, respectively) than the LPS-CON group (14.3%). Error bars represent standard deviations.

**Figure 7 animals-15-01149-f007:**
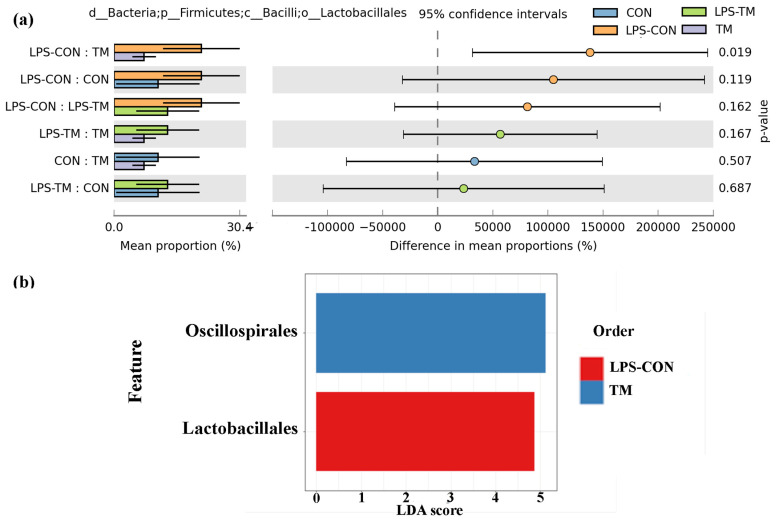
Representation of the relative abundance of *Lactobacillales* and LDA score across experimental groups. (**a**) A statistically significant difference (*p* < 0.019) in *Lactobacillales* abundance was observed between the TM and LPS-CON groups. (**b**) LDA further highlighted the enrichment of *Lactobacillales* in the LPS-CON group and the enrichment of *Oscillospirales* in the TM group.

**Figure 8 animals-15-01149-f008:**
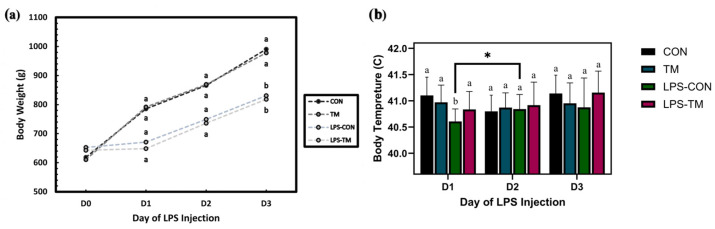
Effect of LPS challenge on body weight (BW) and body temperature (BT) in broiler chickens subjected to temperature manipulation (TM) during embryogenesis. (**a**) LPS significantly reduced BW (*p* < 0.001), while TM did not significantly affect BW (*p* > 0.05). (**b**) In the LPS-CON group, BT initially decreased before increasing significantly (*p* = 0.013). In contrast, the LPS-TM group exhibited a non-significant increase in body temperature over time (*p* > 0.05). Groups labeled with different letters (^a, b^) differ significantly within the same day (*p* < 0.05); asterisks (*) indicate significant differences between days.

**Table 1 animals-15-01149-t001:** Experimental diet composition based on NRC (1994) guidelines.

Ingredient (% of Diet)	Starter	Grower
Corn	56.80	64.76
Soybean Meal (CP 44%)	35.40	27.85
Fish Meal	1.00	1.00
Soybean Oil	2.31	1.39
Oyster Shell	1.34	1.84
Dicalcium Phosphate	1.53	1.66
Common Salt	0.396	0.326
Vitamin Premix ^a^	0.50	0.50
Mineral Premix ^b^	0.50	0.50
DL-Methionine	0.151	0.055
L-Lysine HCl	0.073	0.119
Total	100	100
Nutrient Composition		
Nutrient	Starter	Grower
AMEn (Kcal/Kg)	2950	2960
Crude Protein (%)	21.203	18.499
Arg (%)	1.365	1.157
Lys (%)	1.208	1.052
Met (%)	0.490	0.360
Met + Cys (%)	0.832	0.666
Ca (%)	0.997	1.200
Available *p* (%)	0.453	0.467
Na (%)	0.180	0.150

^a^ Vitamin Mix (per kilogram of food): Calcium Pantothenate, 19.6 mg; Niacin, 59.4 mg; Pyridoxine, 5.88 mg; Folic Acid, 2 mg; Vitamin B_12_, 0.03 mg; Biotin, 0.2 mg; Choline Chloride, 500 mg; Antioxidant, 2 mg; Vitamin A, 18,000 IU; Vitamin D_3_, 4000 IU; Vitamin E, 36 mg; Vitamin K_3_, 4 mg; Thiamine, 3.5 mg; Riboflavin, 13.2 mg. ^b^ Mineral Premix (per kilogram of food): 198.4 mg of Mn, 169.4 mg of Zn, 100 mg of Fe, 20 mg of Cu, 1.98 mg of I, and 0.4 mg of Se.

**Table 2 animals-15-01149-t002:** Alpha diversity indexes of microbial communities.

Index	Con	TM	LPS-CON	LPS-TM	*p*-Value
Chao1	211.56 ± 41.6	188.5 ± 47.2	197.23 ± 62.96	222.1 ± 37.97	0.84
Shannon	3.75 ± 0.25	3.76 ± 0.31	3.77 ± 0.54	3.91 ± 0.34	0.43

Abbreviations: CON = control; TM = thermal manipulation; LPS-CON = control with lipopolysaccharide challenge; LPS-TM = thermal manipulation with lipopolysaccharide challenge. Values represent mean ± SD (*n* = 6). No statistically significant differences were observed between groups (*p* > 0.05; two-way ANOVA).

**Table 3 animals-15-01149-t003:** Summary of PERMANOVA results for microbial community composition.

Pair	F-Value	*p*-Value
TM vs. CON	2.033	0.041
TM vs. LPS-CON	3.4913	0.012
TM vs. LPS-TM	1.5514	0.08
CON vs. LPS-CON	1.0491	0.398
CON vs. LPS-TM	1.3863	0.143
LPS-CON vs. LPS-TM	1.6511	0.087

## Data Availability

All raw sequence data (FASTQ files) and QIIME 2 artifacts, including quality control files, feature tables, taxonomy tables, and rooted phylogenetic trees, associated with this study are publicly available and can be accessed through the following link: https://justedujo-my.sharepoint.com/:f:/g/personal/mbvlab_just_edu_jo/EhFXpakHEJhPgF-CzA2CqSEB0pvha6zKKggahhonMlQE0g?e=QsyF7j (accessed on 9 April 2025). Additionally, the raw sequence data are also available from the NCBI SRA database under accession number PRJNA1249260: https://www.ncbi.nlm.nih.gov/bioproject/1249260 (accessed on 26 March 2025).

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
