# Peer review of "Effects of Embryonic Thermal Manipulation on Body Performance and Cecum Microbiome in Broiler Chickens Following a Post-Hatch Lipopolysaccharide Challenge"

_animals, 2025, doi:10.3390/ani15081149_

Round 1
Reviewer 1 Report
Comments and Suggestions for Authors
- Were the raw reads normalized before making any comparisons and drawing conclusions for the microbiome study?
- After the hatch, Was there any change in the body weight of the birds between thermal manipulation and after thermal manipulation? I am curious to know the changes that might be there? May be there is change in immune and microbial response itself.
- Looking at the data, it seems like there was no dramatic reduction in body weight with LPS-challenged birds irrespective of TM. I was wondering why LPS challenge was considered on day 19 instead of day 7 of post-hatch.
- It will be nice to look at the changes in microbiome in early days after TM before performing experiment to LPS challenge birds in PH day 19.
- I was wondering why the study was carried out just for 4 days instead of a week or two following the challenge to LPS. What is the logic behind having such an experimental design.
- I do not agree with the statement mentioned in lines 356-357. As I mentioned earlier in my comments, experimental design is not rigorous enough to make such a conclusion. There are not enough results to make such conclusions.
Author Response
|
Comments 1: Were the raw reads normalized before making any comparisons and drawing conclusions for the microbiome study?
|
|
Response 1: Thank you for your insightful comment. We acknowledge the importance of ensuring proper normalization before conducting microbiome analyses to avoid biases that could affect our conclusions.…. Yes, the raw reads were normalized before making comparisons and drawing conclusions for this study. To account for variations in sequencing depth, rarefaction was performed at a depth of 19,940 reads per sample. This method standardizes library sizes across all samples, ensuring comparability in alpha- and beta-diversity analyses. While rarefaction is a widely accepted approach to mitigate differences in sequencing depth, we recognize the ongoing debate regarding its potential drawbacks, such as loss of data from low-abundance taxa. However, in our case, rarefying the data was necessary to maintain uniform sequencing coverage across groups. Additionally, Hellinger transformation was applied to standardize data for Canonical Correspondence Analysis (CCA). This transformation provides the highest accuracy for unfiltered data. Furthermore, we have now explicitly stated in the revised manuscript (Page 7, Lines 243–244) that: "Based on the rarefied feature table, differential abundance testing was conducted using LEfSe (Linear Discriminant Analysis Effect Size) and the Kruskal-Wallis H-test." This clarification ensures transparency in the statistical methods applied for microbiome analysis. |
|
Comments 2: After the hatch, Was there any change in the body weight of the birds between thermal manipulation and after thermal manipulation? I am curious to know the changes that might be there? May be there is change in immune and microbial response itself. |
|
Response 2: Thank you for your thoughtful question. We recorded body weight throughout the entire rearing period and found that TM increased BW only at day 5 of age compared to the control group. However, this effect did not persist later in the experiment.
Influence of TM on Post-Hatch Growth: Previous research has shown that TM can influence post-hatch growth through metabolic and muscular adaptations, but the effects vary depending on: Ø Incubation conditions (e.g., temperature intensity, duration, and timing of TM). Ø Genetic strain of the birds, which can affect their metabolic and physiological responses to TM. However, literature on the long-term effects of TM on BW remains mixed. Some studies report increased BW, while others find no significant impact or even reductions in growth under certain TM protocols.
Study Focus and Justification: Since this study primarily investigated LPS challenge and microbial responses, the focus was not on TM’s long-term effects on growth performance. Instead, we aimed to assess whether TM could modulate gut microbiota during an inflammatory challenge. |
|
Comments 3: Looking at the data, it seems like there was no dramatic reduction in body weight with LPS-challenged birds irrespective of TM. I was wondering why LPS challenge was considered on day 19 instead of day 7 of post-hatch. |
|
Response 3: The selected LPS dosage (0.5 mg/kg BW), timing (day19) and three-consecutive-day injection protocol were based on established models from peer-reviewed studies (Wu, Li [35]; Zheng, Zhang [36]) that demonstrated:
Ø Consistent villus damage Ø Affect inflammatory cytokines production Ø Affect microbiome and lead to dysbacteriosis
Ø Cumulative stress response mimicking natural disease progression Ø Reproducible morphological changes (confirmed by histopathology) Ø Sustained oxidative stress markers Accordingly, we have updated the Materials and Methods section in the revised manuscript (Page 5, Lines 156–164).
Additionally, our data do in fact show significant growth impacts from LPS challenge (P<0.05), with clear weight reduction: Ø LPS groups (LPS-CON/LPS-TM) showed ~17% lower final weights (818-830g) vs controls (991g) Ø Statistically distinct groupings (marked "a" vs "b") confirm significance difference
|
|
Comments 4: It will be nice to look at the changes in microbiome in early days after TM before performing experiment to LPS challenge birds in PH day 19. |
|
Response 4: We sincerely appreciate the reviewer's thoughtful suggestions regarding study limitations. In response, we have added a dedicated "Study Limitations" section (Page 15, Lines 557–565) to the manuscript that addresses two key aspects: Ø The male-only study design, where we acknowledge that sex-specific microbiome differences in broilers may affect the generalizability of our findings due to budget constraints.
Ø The importance of characterizing early post-hatch microbiome dynamics (days 1-18) prior to LPS challenge to better understand how TM shapes initial microbial colonization patterns.
|
|
Comments 5: I was wondering why the study was carried out just for 4 days instead of a week or two following the challenge to LPS. What is the logic behind having such an experimental design. |
|
Response 5: We thank the reviewer for this important question regarding our experimental timeline. The 4-day study duration following LPS challenge was carefully selected based on several biological and methodological considerations: I. Acute Phase Focus: The inflammatory and oxidative stress responses to LPS peak within 48-72 hours post-injection, as demonstrated by Zheng and Zhang [36] in their comparable broiler study using injections at 16, 18 and 20 days. Our design specifically targets this critical window to capture the primary effects of LPS challenge before compensatory mechanisms emerge. II. Prevention of Confounding Factors: Extending the observation period beyond 4 days risks introducing secondary adaptations (e.g., recovery adaptation, microbial community shifts, compensatory growth) that could obscure the direct effects of LPS and TM we aimed to study. III. Animal Welfare Considerations: The 4-day period allowed us to assess acute responses while minimizing prolonged stress on the birds, in line with ethical guidelines for challenge studies.
|
|
Comments 6: I do not agree with the statement mentioned in lines 356-357. As I mentioned earlier in my comments, experimental design is not rigorous enough to make such a conclusion. There are not enough results to make such conclusions. |
|
Response 6: Thank you for your feedback. We acknowledge your concern regarding the strength of our conclusions in lines 356-357. To address this, we have expanded our discussion (page 12, lines 409-431) to provide additional context and supporting evidence for our hypothesis.
Furthermore, we have explicitly disclosed the study's limitations, recognizing that while our findings suggest a potential relationship between TM, gut microbiota, and immune modulation, additional studies with a more rigorous experimental design (e.g., multi-timepoint microbiome analysis, and functional metagenomics) are necessary to confirm these effects.
We appreciate your constructive comments and believe these revisions will enhance the clarity and scientific robustness of our conclusions.
|

Reviewer 2 Report
Comments and Suggestions for Authors
Line 54-58: What's meaning of the word “it”. Rephrase this sentence.
Line 72-75: provide some evidences about the relationship between LPS and TM.
Line 93: The committee name don't abbreviate.
Line 94: (347/12/4/16). what’s the meaning.
Line 103: Why chose the seventh day of incubation?
Line 105: Why not directly choose eggs of similar weight on the seventh day of incubation.
Line 108: Why the embryonic days 10-18 was excepted.
Figure 1: there are too many point-in-time, the experiment design is unclear. The point-in-time set by TM and LPS challenge is enough.
Line 356-391: This part is not discussion, it is only introduction of methods.
In the Materials and Methods, introduction of detection method take up a lot of space, Please simplify this part.
In the discussion, about LPS challenge and TM, the author describes many beneficial effects including muscle development, SCFA-producing, et al. However, author do not verify any indicator, even muscle mass and yield. Why do not that.
The analysis of 16S RNA was pivotal result in this study, but there is too little discussion about the function of the microbiota. Most of the content of discussion is about to study the effects of LPS and TM on signal pathways. Please revised.
Figure 8 is very dim, and the edge of the picture was included in the screenshot. Unacceptable.
Figure 5, 6, and 7 is very dim.
The number of references is too much for a research paper.
Comments on the Quality of English LanguageThe writing unacceptable, these are some examples
Line 32: Change “offering” to “ which offering”
Line 69: The meaning of word “linked” is inappropriate. Change this word.
Line 98: Change “summarizes” to “summarized”.
Line 101: delete “Abnormal eggs, extremely light (less than 45g), and heavy (more than 65g) were excluded”.
Line 119: animal house at Jordan University of Science and Technology.
Line 120: “broiler males” was wrong.
Line 135: Do not need reference.
Line 131-133: The meaning of this sentence was consistent with Table 1. Delete.
......
Please check and revise the whole manuscript.
Author Response
|
Comments 1: Line 54-58: What's meaning of the word “it”. Rephrase this sentence.
|
|
Response 1: We appreciate the reviewer's helpful comment regarding pronoun clarity. We have revised the sentence to eliminate ambiguity and improve readability. The updated text now reads: Chickens are essential for meeting the rising global demand for animal protein. Specifically, in 2021, chicken meat was not only the most produced meat worldwide, but also reached approximately 120 million tons |
|
Comments 2: Line 72-75: provide some evidences about the relationship between LPS and TM.
|
|
Response 2: We sincerely appreciate your insightful suggestion regarding the LPS-TM relationship. Your comment has significantly strengthened our mechanistic discussion. Below we provide supporting references for this key interaction, and we have incorporated these modifications in the revised manuscript (Page 2, Lines 77–81):
The protective effects of TM against LPS-induced inflammation are well-documented in poultry studies: Ø Cytokine Modulation: · Previous research has demonstrated that TM influences immune responses by modulating splenic cytokine expression and heat shock protein (HSP) mRNA levels in LPS-challenged Pekin duck and broiler chickens.
Ø Heat Shock Protein Response: · Previous research has demonstrated that TM modulating heat shock protein (HSP) mRNA expression in LPS-challenged Pekin duck and broiler chickens.
|
|
Comments 3: Line 93: The committee name don't abbreviate. Line 94: (347/12/4/16). what’s the meaning. |
|
Response 3: We sincerely appreciate this careful observation. We have revised the text to spell out the full committee’s name as: "The Animal Care and Use Committee of Jordan University of Science and Technology (Approval Number: 347/12/4/16) reviewed and approved all experimental procedures." |
|
Comments 4: Line 103: Why chose the seventh day of incubation?
|
|
Response 4: ED7 candling was selected because: (1) vascular development enables reliable viability assessment, (2) captures the early embryonic mortality, and (3) follows established TM protocols
|
|
Comments 5: Line 105: Why not directly choose eggs of similar weight on the seventh day of incubation. |
|
Response 5: We appreciate the reviewer's careful reading and this important methodological question. Upon re-examining our manuscript, we recognize the text inadvertently suggested eggs were divided after candling, when in fact the grouping occurred before incubation. The complete experimental sequence was:
We have revised the Methods section to clarify this sequence (page 3 Lines 110-118)
|
|
Comments 6: Line 108: Why the embryonic days 10-18 was excepted. |
|
Response 6: We thank the reviewer for this valuable suggestion.. We provided a scientific rationale based on previous studies showing that these TM parameters result in changes to the expression of heat shock proteins and inflammatory cytokines, as well as a reduction in the expression of genes linked to oxidative stress and an increase in antioxidant capacity. Accordingly, we have updated the Materials and Methods section in the revised manuscript (Page 3, Lines 119–122). |
|
Comments 7: Figure 1: there are too many point-in-time, the experiment design is unclear. The point-in-time set by TM and LPS challenge is enough. |
|
Response 7: We appreciate the reviewer’s feedback regarding the complexity of our experimental timeline in Figure 1. We would like to clarify that the multiple time points were intentionally included to demonstrate methodological rigor:
Ø Showing the full experimental workflow (from egg selection to LPS challenge) ensures reproducibility, which is especially important for TM studies where treatment conditions significantly impact results.
Ø Post-hatch LPS timing is retained to show the connection between embryonic TM and later immune challenges.
|
|
Comments 8: Line 356-391: This part is not discussion, it is only introduction of methods.
|
|
Response 8: We thank the reviewer for this valuable suggestion.. The methodological content originally appearing in the Discussion section (lines 356-391) has been relocated to its proper position in the Materials and Methods section (now page 6, lines 178-184).
|
|
Comments 9: In the Materials and Methods, introduction of detection method take up a lot of space, Please simplify this part. |
|
Response 9: We agree that methodological details can be streamlined in the main text. The revised version focuses on key workflows, while Supplementary Material S1 provides intensive protocols for reproducibility. This balances readability with technical rigor. |
|
Comments 10: In the discussion, about LPS challenge and TM, the author describes many beneficial effects including muscle development, SCFA-producing, et al. However, author do not verify any indicator, even muscle mass and yield. Why do not that. |
|
Response 10: We sincerely appreciate the reviewer’s valuable comment. The study primarily focused on establishing the impact of TM on microbial resilience to LPS challenge, with observed shifts in SCFA-producing taxa suggesting potential metabolic benefits. However, we acknowledge that direct measurements of muscle mass, yield, or SCFA concentrations were not included in this experimental design.
Limitations Added to (page 15, lines 534-539): "Although our findings suggest a potential link between TM, immune modulation, and metabolic efficiency, we did not directly measure muscle mass, carcass yield, or SCFA production. Future studies should incorporate muscle fiber histology, carcass composition analysis, metabolomic profiling, multi-timepoint microbiome analysis, and functional metagenomics to establish a more direct relationship between TM, microbial modulation, and overall broiler performance."
|
|
Comments 11: The analysis of 16S RNA was pivotal result in this study, but there is too little discussion about the function of the microbiota. Most of the content of discussion is about to study the effects of LPS and TM on signal pathways. Please revised. |
|
Response 11: We thank the reviewer for this insightful suggestion. In the revised manuscript, we have significantly expanded the functional interpretation of microbiome findings to better highlight the microbiota's ecological and metabolic roles:
a) Oscillospirales Functional Significance (Page 13, Lines 408–424)
b) Lactobacillales Functional Significance (Page 13, Lines 435–446)
Ø Added their established correlation with SCFA production and CD4+ T-cell modulation to explain immune homeostasis maintenance
Ø Clarified lactic acid's dual function in both pH reduction and direct pathogen membrane disruption
Ø Explicitly framed their expansion as a protective host-microbiota adaptation against inflammatory damage
Ø Cited Alaqil demonstrating L. acidophilus' stress-alleviating effects and HSP70 reduction in LPS-challenged hens
|
|
Comments 13: Figure 8 is very dim, and the edge of the picture was included in the screenshot. Unacceptable. Figure 5, 6, and 7 is very dim.
|
|
Response 13: Thank you for your feedback regarding Figures 5, 6, 7, and 8. We acknowledge that the figures may appear dim in the PDF version due to size reduction during the submission process. However, we confirm that high-resolution, full-size images were uploaded separately to the journal submission system, as per the journal's guidelines. The original figures should be clear when viewed at full resolution. Additionally, for Figure 8, we have removed the edge of the picture to ensure clarity and proper formatting. If necessary, we are happy to re-upload the figures or provide alternative formats to ensure clarity. Please let us know if further adjustments are required. |
|
Comments 14: The number of references is too much for a research paper. |
|
Response 14: We sincerely appreciate your feedback regarding the number of references in our manuscript. We have carefully reviewed the reference list to ensure it remains focused and relevant while adhering to scholarly standards.
Actions Taken:
· Redundant references not directly supporting key findings have been removed.
· Priority was given to recent, high-impact citations to strengthen the manuscript’s credibility.
As per the journal’s guidelines, references cited in the supplementary material are included in the main reference section. However, if the journal enforces a strict reference limit, we are happy to further refine the list upon request. |
|
4. Response to Comments on the Quality of English Language |
|
Point 1: Line 32: Change “offering” to “which offering” |
|
Response 1: Thank you for your suggestion regarding Line 32. We have revised the phrase to 'which offers' for grammatical accuracy, as 'which' requires a conjugated verb. The sentence now reads: “Thermal manipulation (TM) during embryogenesis has emerged as a promising strategy to enhance post-hatch performance and improve resilience to environmental and bacterial stress, which offers a potential alternative to reduce the reliance on antibiotic growth promoters (AGPs) in broiler production” |
|
Point 2: Line 69: The meaning of word “linked” is inappropriate. Change this word. |
|
Response 2: I appreciate your comment about Line 69. We modified the sentence to make it more grammatically correct. “Studies demonstrate that TM promotes muscle growth, strengthens resistance to bacterial challenges, and enhances immune responses under heat stress” |
|
Line 98: Change “summarizes” to “summarized”. Line 101: delete “Abnormal eggs, extremely light (less than 45g), and heavy (more than 65g) were excluded”. Line 119: animal house at Jordan University of Science and Technology. Line 120: “broiler males” was wrong. Line 135: Do not need reference. Line 131-133: The meaning of this sentence was consistent with Table 1. Delete. |
|
Response: We sincerely appreciate your thorough review and valuable suggestions. The following revisions have been made to address your comments:
|

Reviewer 3 Report
Comments and Suggestions for Authors
This research aimed to investigate how embryonic thermal manipulation (TM)—a method adjusting incubation temperature and humidity—influences broiler chicken resilience to post-hatch lipopolysaccharide (LPS)-induced stress by examining growth performance, body temperature, and gut microbiota. The control group (CON) was incubated at standard conditions (37.8°C, 56% RH), while the TM group underwent higher temperature and humidity (39°C, 65% RH) daily for 18 hours during embryonic days 10–18. Post-hatch, both groups were challenged with LPS injections to induce stress. The research concluded that embryonic TM can beneficially modulate gut microbiota, potentially enhancing broiler resilience against immunological stressors like LPS. However, TM's role in maintaining body weight during inflammation remains uncertain, warranting further optimization and research.
The manuscript strongly aligns with the scope and interests of Animals, addressing critical areas of poultry health, antibiotic reduction, and microbiome interactions under stress conditions. Despite strong scientific merit and relevance, significant improvements in methodological clarity, scientific interpretation, and detailed discussions of physiological implications and practical applicability are necessary before the manuscript can be considered for publication. Below are detailed comments and suggestions for improvement.
Simple Summary and Abstract:
Line 24-27: Clearly specify which bacterial taxa most prominently benefited from TM to provide precise information early in the abstract.
Introduction:
Line 78-84: Expand the literature review by briefly mentioning prior research gaps explicitly—e.g., clarify why TM effects on microbiota under stress conditions remain underexplored.
Materials and Methods:
Line 104-110 (Experimental design): Clearly justify why TM conditions (39°C, 65% RH, 18 hours) were chosen based on previous studies or preliminary findings?
Line 143-147 (LPS challenge): Specify the rationale for the chosen LPS dosage and the number of injections (0.5 mg/kg, three consecutive days).
Results:
Line 285-286 (Table 3?): Check the table name; it should be Table 3 Pairwise PERMANOVA analysis.
Discussion:
Line 392-396: Expand upon potential mechanisms by which TM affects gut microbiota composition. Specifically, include physiological or epigenetic mechanisms linking thermal manipulation and microbiota shifts.
Line 414-419: Clearly state the implications of Oscillospirales enrichment for poultry gut health, including known functions or roles in nutrient metabolism.
Line 428-437 (Lactobacillales): Further discuss why Lactobacillales might increase specifically under LPS stress conditions—clarify competitive advantages or immune-modulatory roles.
Line 452-455 (male-only study): Explicitly discuss limitations of only including males, particularly concerning microbiota composition differences between sexes.
Line 452-455: Explicitly discuss how TM might be scaled up in practical poultry operations, mentioning potential challenges and economic considerations.
Conclusion:
Recommend adding practical implications for the poultry industry, particularly regarding TM protocol implementation at scale.
References:
Please carefully check all in-text citations and the reference list at the end of the manuscript to ensure they conform precisely to the citation format required by Animals (MDPI). Pay particular attention to citation numbering, punctuation, author initials, journal abbreviations, volume and page formatting, and URL inclusion to align fully with MDPI guidelines.
Author Response
|
Comments 1: Simple Summary and Abstract:
Line 24-27: Clearly specify which bacterial taxa most prominently benefited from TM to provide precise information early in the abstract.
|
|
Response 1: We have explicitly stated that TM had a different composition of gut bacteria, particularly by increasing beneficial bacterial taxa such as Oscillospirales. Thank you for pointing this out. We agree with this comment. Therefore, we have…. Updated the Abstract and Simple Summary to include these specific taxa (page 1, lines 25-29). |
|
Comments 2: Introduction: Line 78-84: Expand the literature review by briefly mentioning prior research gaps explicitly—e.g., clarify why TM effects on microbiota under stress conditions remain underexplored. |
|
Response 2: Agree. We thank the reviewer for this constructive suggestion. We have revised the Introduction to clearly highlight prior research gaps regarding TM's effects on microbiota under stress conditions. The changes can be found in the revised manuscript on: Page2, (Lines 80-90) of the revised manuscript. Changes Made: Added clear discussion of gaps in TM-microbiota-LPS research Incorporated supporting references Enhanced the rationale for investigating TM's potential to modulate microbial resilience against LPS-induced gastrointestinal stress |
|
Comments 3: Materials and Methods:
Line 104-110 (Experimental design): Clearly justify why TM conditions (39°C, 65% RH, 18 hours) were chosen based on previous studies or preliminary findings? |
|
Response 3: We provided a scientific rationale based on previous studies showing that these TM parameters result in changes to the expression of heat shock proteins and inflammatory cytokines, as well as a reduction in the expression of genes linked to oxidative stress and an increase in antioxidant capacity. Accordingly, we have updated the Materials and Methods section in the revised manuscript (Page 3, Lines 116–119). |
|
Comments 4: Materials and Methods:
Line 143-147 (LPS challenge): Specify the rationale for the chosen LPS dosage and the number of injections (0.5 mg/kg, three consecutive days). |
|
Response 4: The selected LPS dosage (0.5 mg/kg BW) and three-consecutive-day injection protocol were based on established models from peer-reviewed studies (Wu et al. [35]; Zheng et al. [36]) that demonstrated: I. Dose Validation: 0.5 mg/kg effectively induces intestinal inflammation without causing excessive mortality, as shown by: Ø Consistent villus damage Ø Affect inflammatory cytokines production Ø Affect microbiome and lead to dysbacteriosis II. Protocol Optimization: Three consecutive injections produced: Ø Cumulative stress response mimicking natural disease progression Ø Reproducible morphological changes (confirmed by histopathology) Ø Sustained oxidative stress markers Accordingly, we have updated the Materials and Methods section in the revised manuscript (Page 5, Lines 156–164). |
|
Comments 5: Results:
Line 285-286 (Table 3?): Check the table name; it should be Table 3 Pairwise PERMANOVA analysis. |
|
Response 5: We sincerely appreciate the reviewer's careful attention to detail. The table reference has been corrected to Table 3. |
|
Comments 6: Discussion:
Line 392-396: Expand upon potential mechanisms by which TM affects gut microbiota composition. Specifically, include physiological or epigenetic mechanisms linking thermal manipulation and microbiota shifts. |
|
Response 6: We thank the reviewer for this valuable suggestion.. In the revised manuscript (Page 12, Lines 409–431), we have expanded the discussion of TM’s effects on gut microbiota to include: I. Epigenetic mechanisms: Added evidence that TM alters histone modifications (H3K4me3, H3K27me3) in embryonic tissues, which may extend to gut epithelium
II. Critical windows: Highlighted the peri-hatching period as pivotal for microbiota establishment
III. Comparative evidence: Incorporated parallel findings from probiotic studies showing microbiota-epigenome crosstalk |
|
Comments 7: Discussion:
Line 414-419: Clearly state the implications of Oscillospirales enrichment for poultry gut health, including known functions or roles in nutrient metabolism. |
|
Response 7: We thank the reviewer for this valuable suggestion.. In the revised manuscript (Page 13, Lines 449–468), we have clearly highlighted the implications of Oscillospirales enrichment for poultry gut health by: I. Nutrient Metabolism: Emphasizing their role in fermenting indigestible fibers/starch to SCFAs (propionate, valerate, butyrate) that enhance energy harvest
II. Gut Barrier Function: Detailing butyrate’s dual role as both an enterocyte energy source and regulator of epithelial integrity
III. Health Impacts: Connecting histone hyperacetylation modulated by butyrate to practical outcomes (reduced inflammation, improved gut morphology) relevant to poultry production. |
|
Comments 8: Discussion:
Line 428-437 (Lactobacillales): Further discuss why Lactobacillales might increase specifically under LPS stress conditions—clarify competitive advantages or immune-modulatory roles. |
|
Response 8: We thank the reviewer for this valuable suggestion.. In the revised manuscript (Page 13, Lines 476–487), Our discussion of Lactobacillales' LPS response has been strengthened by:
I. Immunomodulatory Role Ø Added their established correlation with SCFA production and CD4+ T-cell modulation to explain immune homeostasis maintenance
II. Competitive Advantage Ø Clarified lactic acid's dual function in both pH reduction and direct pathogen membrane disruption
III. Host-Adaptive Response Ø Explicitly framed their expansion as a protective host-microbiota adaptation against inflammatory damage
Ø Cited Alaqil demonstrating L. acidophilus' stress-alleviating effects and HSP70 reduction in LPS-challenged hens
|
|
Comments 9: Discussion:
Line 452-455 (male-only study): Explicitly discuss limitations of only including males, particularly concerning microbiota composition differences between sexes. |
|
Response 9: We appreciate the reviewer's valuable comment regarding sex-specific considerations.. In response, we have added a dedicated "Study Limitations" section (Section 6, page 15, lines 557-565) acknowledging that:
I. Our exclusive use of male chicks was due to budget constraints
II. This design choice may affect generalizability given known sex differences in broiler microbiomes
III. Future studies should include both sexes to account for sex-specific variations in microbial composition and function
|
|
Comments 10: Conclusion:
Recommend adding practical implications for the poultry industry, particularly regarding TM protocol implementation at scale. |
|
Response 10: We thank the reviewer for this valuable suggestion regarding practical applications... In response, we have strengthened the conclusion by:
I. Explicitly positioning TM as a viable antibiotic-free production strategy
II. Outlining key research priorities for commercial implementation (strain-specific optimization, probiotic integration)
III. Highlighting the potential hatchery-level benefits (flock health improvement, antibiotic reduction) |
|
Comments 11: References:
Please carefully check all in-text citations and the reference list at the end of the manuscript to ensure they conform precisely to the citation format required by Animals (MDPI). Pay particular attention to citation numbering, punctuation, author initials, journal abbreviations, volume and page formatting, and URL inclusion to align fully with MDPI guidelines. |
|
Response 11: We sincerely appreciate the reviewer’s careful attention to reference formatting.... We have carefully reviewed and standardized all in-text citations and reference list entries to fully comply with Animals (MDPI) formatting guidelines. |

Round 2
Reviewer 2 Report
Comments and Suggestions for Authors No further comments Best Regards Comments on the Quality of English Languageplease check the whole manuscript about grammar problems
Reviewer 3 Report
Comments and Suggestions for Authors
Accept in present form